# Insulated transcriptional elements enable precise design of genetic circuits

Yeqing Zong[1,2], Haoqian M. Zhang [3,4], Cheng Lyu[5], Xiangyu Ji[1,2], Junran Hou[1,2], Xian Guo[1,2], Qi Ouyang[3,4,5] & Chunbo Lou[1,2]

Rational engineering of biological systems is often complicated by the complex but unwanted interactions between cellular components at multiple levels. Here we address this issue at the level of prokaryotic transcription by insulating minimal promoters and operators to prevent their interaction and enable the biophysical modeling of synthetic transcription without free parameters. This approach allows genetic circuit design with extraordinary precision and diversity, and consequently simplifies the design-build-test-learn cycle of circuit engineering to a mix-and-match workflow. As a demonstration, combinatorial promoters encoding NOT-gate functions were designed from scratch with mean errors of <1.5-fold and a success rate of >96% using our insulated transcription elements. Furthermore, four-node transcriptional networks with incoherent feed-forward loops that execute stripe-forming functions were obtained without any trial-and-error work. This insulation-based engineering strategy improves the resolution of genetic circuit technology and provides a simple approach for designing genetic circuits for systems and synthetic biology.

[1] Key Laboratory of Microbial Physiological and Metabolic Engineering, Institute of Microbiology, Chinese Academy of Sciences, Beijing 100101, China. [2] College of Life Sciences, University of Chinese Academy of Sciences, Beijing 100049, China. [3] Peking-Tsinghua Joint Center for Life Sciences, Peking University, Beijing 100871, China. [4] Center for Quantitative Biology, Peking University, Beijing 100871, China. [5] School of Physics, Peking University, Beijing 100871, China. Yeqing Zong and Haoqian M. Zhang contributed equally to this work. Correspondence and requests for materials should be addressed to Q.O. (email: qi@pku.edu.cn) or to C.L. (email: louchunbo@im.ac.cn)

Biological systems such as living cells are characterized by highly complex interactions between numerous genetic and cellular components[1]. Correspondingly, the engineering of biological systems—a process used to adapt biological components from nature and apply them in novel design combinations—often suffers from unwanted complexity at various biological levels[2]. Therefore, in order to enhance the reliability and predictability of biological systems engineering, this complexity must be managed or reduced[3]. Synthetic biology seeks to address this issue by developing foundational theories and technologies at the systems level[4–6] using various approaches, such as developing design strategies for synthetic genetic circuits based on principles of mature engineering disciplines[7, 8], reverse engineering of natural genetic circuits to create their synthetic counterparts[9–11], and exploration of design constraints for dealing with the complex interactions between genetic circuits and their biochemical, host, and environmental factors[12–14]. Developments have also occurred at the biological parts and modules level and include composable parts that regulate gene expression with high orthogonality and have minimal interferences with the host cells[15–18], biological insulators that eliminate or buffer against unexpected interferences at the functional[19] and physical[20–23] interfaces of parts/modules, and a corresponding programming environment that supports automated, high-throughput composing of parts/modules for non-experts[24].

Despite these abovementioned developments that enable composing biological parts to build circuits by managing or reducing the systems complexity, methods that enable the rational design of individual basic parts, such as promoters, a crucial need for understanding and manipulating basic processes in gene expression, remain elusive. This is because the interactions between sub-part components (called "elements" hereafter) are also complex. For instance, prokaryotic promoters constitute a category of biological parts that encode transcriptional control of gene expression and function as key factors in information processing. In general, prokaryotic promoters have a multi-element architecture consisting of a minimal promoter (−10 box, −35 box, and transcription-start site +1, called the "promoter core" hereafter) for transcription initiation and one or several operators for transcription factor binding[25]. Canonical views assume that this architecture is modular and that operators simply acts as docking sites for transcription factors that either enhance or repress transcription initiation at the promoter core[26, 27]. However, studies have revealed that the interplay between sub-promoter elements could be far more complex than hypothesized by this "modular assumption", as the operator sequences alone can significantly perturb the intrinsic activity of promoter cores and vice versa[28–30]. Consequently, despite the intensive efforts focused on interpreting and dissecting these modes of interplays in order to develop a rational basis for promoter engineering, either computationally or experimentally[29, 31–33], the bottom-up design of prokaryotic promoters using sub-promoter elements is still largely an ad hoc exercise, with prokaryotic promoters are generally regarded as functionally "indivisible" parts in genetic circuit engineering.

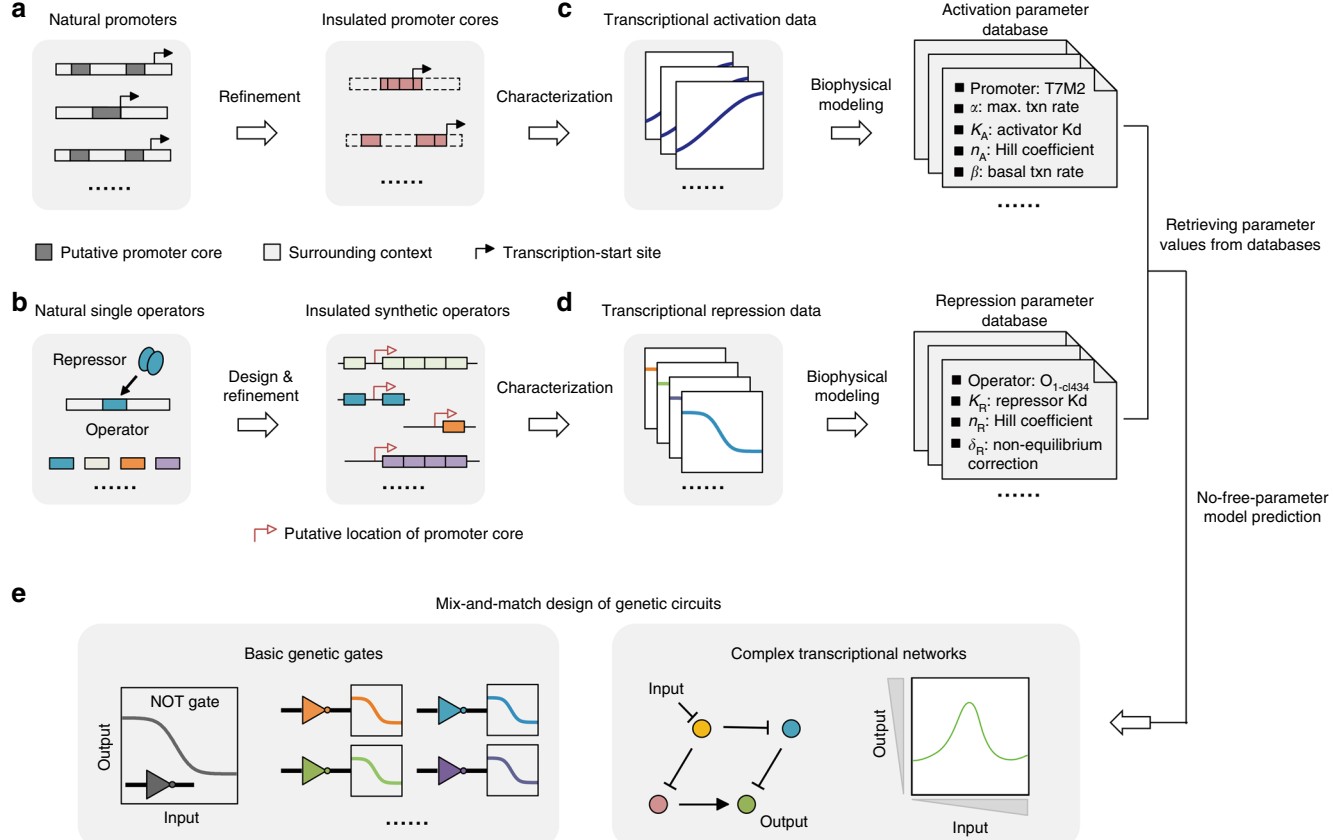

**Fig. 1** Overview of the precise design of genetic circuits using insulated transcriptional elements. **a** Natural promoters from bacteria and phages were genetically refined to identify insulated promoter cores (minimal promoters for which transcriptional activity is independent of sequence context). **b** Natural single operators were permuted and refined to create synthetic operators with diverse transcriptional regulation properties, designed to be free of spontaneous promoter activity. **c** and **d** Insulated promoter cores **c** and synthetic operators **d** were experimentally characterized and parameterized as transcriptional activation- and repression-dependent parameter databases, respectively, via biophysical modeling. **e** Genetic circuits including both basic genetic gates and complex transcriptional networks can be precisely designed using insulated transcriptional elements

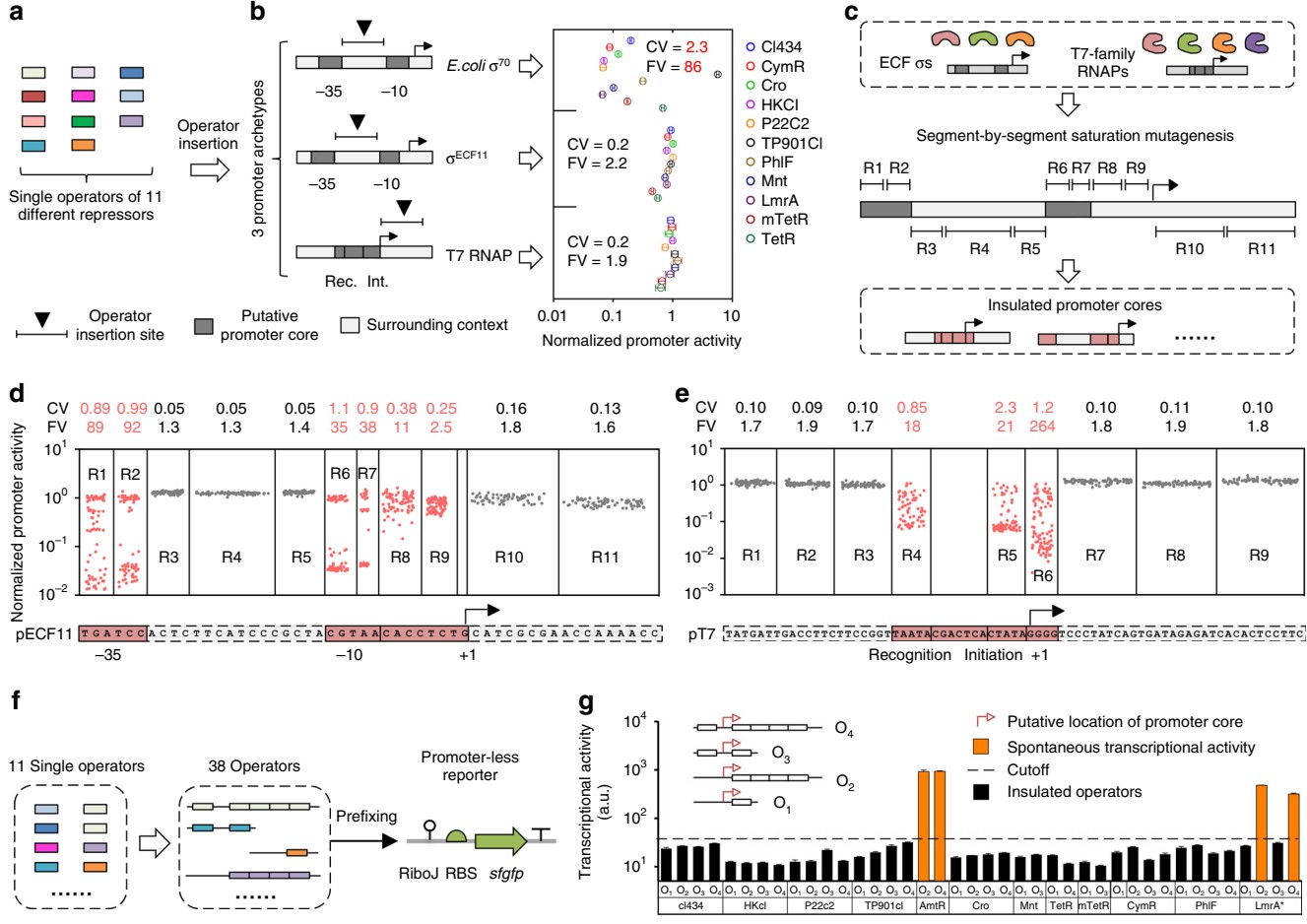

**Fig. 2** Genetic refinement to identify insulated promoter cores and synthetic operators. **a** Single operators targeted by 11 different transcriptional repressors were collected. **b** Operators were individually inserted into the context sequences surrounding the promoter cores of $P_{lac}$ (E. coli $\sigma^{70}$), $P_{ECF11}$ ($\sigma^{ECF11}$), and $P_{T7}$ (T7 RNAP) to evaluate the effect of operator sequences on the transcriptional activity of each promoter core. Data were normalized according to the promoter activity of the respective wild-type promoters. **c** Promoters targeted by ECF σs and T7-family RNAPs were subjected to segment-by-segment saturation mutagenesis to identify the minimal sequence of each promoter core. $P_{ECF11}$ is shown as an example; R1-R11 correspond to different segments (3–9 bp in length) along the promoter sequence. **d** and **e** Effect of random mutagenesis in each promoter segment on the promoter activity of $P_{ECF11}$ **d** and $P_{T7}$ **e**. Each *dot* represents a randomly selected mutant containing mutations in the corresponding segment. Segments with >3-fold variation are highlighted in *red*. **f** Natural single operators were permuted to fit four different promoter architectures and to further analyze their spontaneous transcriptional activity. **g** Spontaneous transcriptional activity of synthetic operators. The cutoff was defined as the threefold of the E. coli background autofluorescence. Promoter activity was measured using superfolder GFP as the reporter and quantified as the arithmetic means of flow cytometry fluorescence data. CV and FV determined by the ratio of maximum promoter activity to the minimum observed value. Data in **b** and **g** represent the means ± SD from at least three replicate experiments conducted on different days; data in **d** and **e** represent the averages. CV, coefficient of variation; FV, relative(-fold) variation; Int., transcriptional initiation site; Rec., recognition site

We reasoned that, if the highly interconnected sub-promoter elements could be decoupled to satisfy the "modular assumption", we would be able to precisely design promoter parts from scratch by simply combining sub-promoter elements. This easy mix-and-match approach would significantly boost our overall capacity for rationally designing genetic circuits. As a demonstration, we randomly combined 53 different promoter cores and 36 different synthetic operators to design 83 combinatorial promoters that encode NOT-gate functions. Furthermore, these insulated transcriptional elements were utilized to build four-node transcriptional networks with incoherent feed-forward loop (IFFL) topology encoding a stripe-forming function. In silico screening of all 30,528 possible designs directly captured the one with the best stripe-forming performance without any trial-and-error work, as verified experimentally (up to 66-fold pulsing, mean errors <1.4-fold for all examined circuit designs). This insulation-based, sub-promoter-resolution circuit engineering methodology provides a mix-and-match approach for the bottom-up design of genetic circuits and will greatly facilitate rational gene expression engineering in synthetic and systems biology.

## Results

**Overview of promoter design.** We first developed a design strategy to functionally insulate the promoter core and surrounding operators and prevent their interaction (Fig. 1). Natural promoters from bacteria and bacteriophages were genetically refined via saturation mutagenesis through the following steps: (i) promoter cores were identified as minimal promoters (DNA sequences <20 bp) that are insensitive to their surrounding sequence context (Fig. 1a); and (ii) synthetic operators used to add transcriptional regulation to these promoter cores were designed using natural operators and further refined to eliminate any unwanted cryptic promoters (Fig. 1b). Due to the modularity

of these refined elements, biophysical models without any free parameter were subsequently derived to parameterize the roles of promoter cores and operators in transcription regulation. In this process, experimental characterization data for individual sub-promoter elements were converted into activation- and repression-dependent parameter databases (Fig. 1c, d). The resulting parameter databases, in conjunction with no-free-parameter modeling were then used for the bottom-up design of promoters with extraordinary feasibility and precision (Fig. 1e). Of these, 80 promoters performed as predicted (mean errors <1.5-fold, $R^2 = 0.95$ across all 1397 predictions), corresponding a success rate of >96%.

**Identification of insulated promoter cores**. Operators are sequence-defined short DNA fragments (usually 10–20 bp) that are responsible for the specific binding of transcription factors. To evaluate how intrinsic transcriptional activity of promoter cores is perturbed by operator identity, we collected 11 single operators recognized by different transcriptional repressors (Fig. 2a; see Supplementary Table 1 for detailed information); the $P_{lac}$ promoter was selected as the archetype of promoter cores recognized by the *Escherichia coli* housekeeping σ factor $σ^{70}$ (Fig. 2b). The collected operators were then individually inserted into the spacer between the −10 and −35 boxes of $P_{lac}$ promoter; this region is reportedly the most often used for operator-mediated transcriptional repression in both natural and synthetic promoters[29]. To exclude the effect of spacer length variation on promoter core activity[25], the operators >18 bp were truncated before insertion to yield a uniform length of 18 bp. Measurements of the activity of the resulting promoter variants revealed an 86-fold variation (CV = 2.3; Fig. 2b), indicating that the $σ^{70}$-dependent promoter core is highly sensitive to the sequence context imposed by the operators. This finding showed that $σ^{70}$-dependent promoter cores are not modular in function and, therefore, are not a good choice for use in the bottom-up design of promoters.

We then turned to other types of promoter cores, such as the Pecf11_3726 promoter ($P_{ECF11}$) recognized by bacterial extracytoplasmic function (ECF) σ factor $σ^{ECF11}$ [34], and the T7 promoter ($P_{T7}$) recognized by T7 RNA polymerase (T7 RNAP; see Supplementary Table 1 for the sequences of promoters, σ factors and RNAPs). Using the method described above, the same set of operators was inserted individually into the spacer region of $P_{ECF11}$ and the immediate downstream region of $P_{T7}$, respectively, to evaluate the effect of operator identity on promoter core activity in pre-built *E. coli* strains constitutively expressing $σ^{ECF11}$ or T7 RNAP. Surprisingly, the resulting $P_{ECF11}$ variants showed only a 2.2-fold variation in transcriptional activity (CV = 0.2); similarly, 1.9-fold variation in transcriptional activity was observed for $P_{T7}$ (CV = 0.2; Fig. 2b), indicating that the promoter cores of $P_{ECF11}$ and $P_{T7}$ are comparatively insensitive to the sequence context imposed by operators.

In order to systematically evaluate the functional modularity of the promoter cores of $P_{ECF11}$ and $P_{T7}$ and to identify their minimal sequences, we divided the promoters and their surrounding sequences into segments of 3–9 bp in length and conducted saturation mutagenesis segment by segment (Fig. 2c). For any given segment, the mutations were introduced by synthesizing the promoter using degenerate primers via PCR annealing. This yielded 11 mutant libraries for $P_{ECF11}$ and 9 mutant libraries for $P_{T7}$. For each mutant library, we tested at least 90 randomly-picked mutants and analyzed the magnitude of their variation in transcriptional activity. The results showed that for $P_{ECF11}$ the segments R1, R2, R6, and R7, which correspond to the −35 and −10 boxes, are crucial for maintaining transcriptional activity (>35-fold variation; CV > 0.89). Segments R8 and R9,

which are located between the −10 box and the transcription-start site, were found to be less crucial but still important (>2.5-fold variation; CV > 0.25; Fig. 2d). To our surprise, the segments R3, R4, R5, R10, and R11, which correspond to the spacer between the −10 and −35 boxes and the sequence immediately downstream of the transcription-start site, were almost totally insensitive to sequence variation, indicating that they are unnecessary for maintaining transcriptional activity (<1.3-fold variation; CV < 0.13; Fig. 2d). Likewise, for $P_{T7}$, the segments corresponding to the promoter core are crucial for promoter activity (>18-fold variation; CV > 0.85). All of the segments outside of the promoter core, however, were found to be dispensable (<1.9-fold variation; CV < 0.11; Fig. 2e). To evaluate whether promoter core modularity is plasmid context-dependent, we performed segment-by-segment mutagenesis of $P_{ECF11}$ on a new plasmid backbone (p15A origin and medium-copy number, in contrast to the previous one of pSC101 origin and low-copy number). The new results were basically the same as those shown in Fig. 2d (Supplementary Fig. 1). Collectively, these results indicate that the promoter cores of $P_{ECF11}$ and $P_{T7}$ are indeed functionally modular, with a minimal length of 19 and 21 bp, respectively.

As promoter recognition stringency is a common feature of ECF σ factors and T7-family RNAPs[16, 34], the cognate promoters of two additional ECF σ factors ($σ^{ECF16}$ and $σ^{ECF20}$) and three T7-family RNAPs (MmP1, T3, and gh-1) were experimentally scanned in the same manner using saturation mutagenesis. The results revealed indispensable promoter cores with dispensable surrounding sequences similar to those of $P_{ECF11}$ and $P_{T7}$ (Supplementary Fig. 2). From the standpoint of engineering, the promoter cores of ECF σ factors and T7-family RNAPs are thus functionally well-insulated, which means that their roles in transcription initiation would be maintained quantitatively, regardless of the operators used in combination with them. Conversely, the context sequences surrounding these promoter cores can serve as ideal locations for operators used to add transcriptional regulation.

**Design and refinement of synthetic operators**. Next, we set out to design synthetic operators using the single operator collection. The synthetic operators would be expected to mediate only transcriptional repression. Therefore, we wanted to detect and exclude operators exhibiting spontaneous transcriptional activity. Four different promoter architectures were utilized in our work, having one or four operators located downstream of a promoter core, with or without one operator in the upstream region (Fig. 2f, g). This was done in order to impart more flexibility and diversity in transcriptional regulation. Architectures with multiple operators would be expected to allow for cooperative binding of transcriptional repressors such as cI homologues, which intrinsically have this ability[35]. The distance between operator centers was set to 10 or multiples of 10 bp in order to maximize the potential for cooperative binding[36]. The single operators from our collection were then permuted to fit these promoter architectures, yielding 40 different synthetic operators that were subsequently connected to a promoter-less reporter cassette (Fig. 2f). Indeed, some synthetic operators, such as $O_{2-LmrA}$ and $O_{4-LmrA}$ bound by the LmrA* repressor, were found to cause considerable reporter expression, up to orders of magnitude above background fluorescence (Fig. 2g). The cryptic $σ^{70}$-dependent promoters, responsible for the unwanted transcriptional activity, were predicted using the BPROM program (http://www.softberry.com/berry.phtml?topic=bprom&group=programs&subgroup=gfindb, Supplementary Table 2). Such operators were consequently excluded from further experiments.

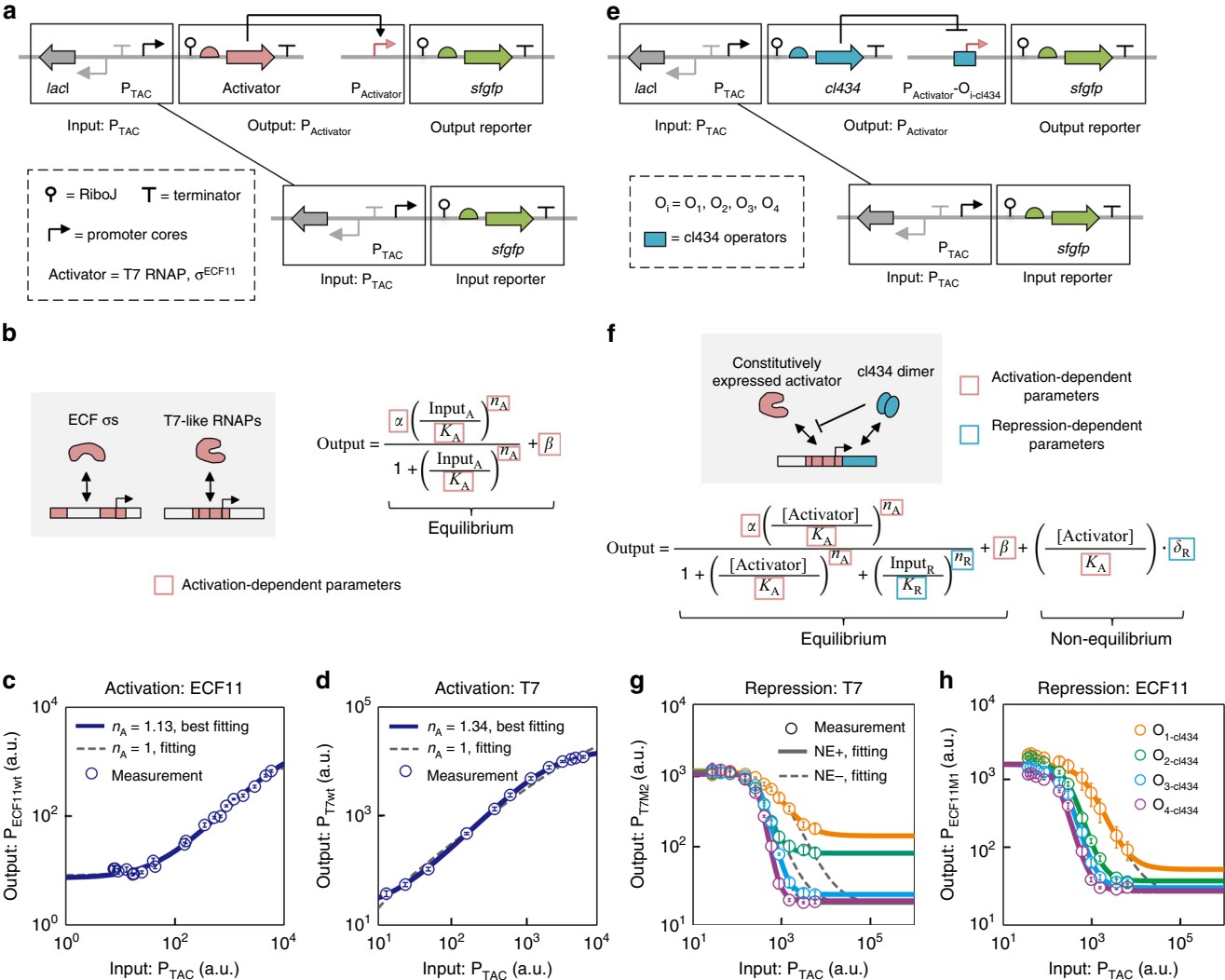

**Fig. 3** Biophysical modeling and parameterization of transcriptional activation and repression. **a** Schematic of genetic circuits used for the characterization of promoter cores. The expression level of each transcriptional activator was determined via fluorescence values of sfGFP transcribed from an independent cassette using the identical input promoter. **b** The biophysical model used to describe the interactions between promoter cores and their cognate transcriptional activators. **c** and **d** Experimental measurements and parameter fitting of response functions for $\sigma^{ECF11}$ **c** and T7 RNAP **d**. *Solid lines* denote parameter fitting when $n_A$ was relaxed; *dashed lines* represent fitting of results with fixed $n_A$ values. **e** Schematic of genetic circuits used for the characterization of repressor operators. The transcriptional activator was constitutively expressed and thus omitted for clarity. The expression level of each transcriptional repressor was determined via fluorescence values of sfGFP transcribed from an independent cassette using the identical input promoter. **f** Biophysical model used to describe the interactions between activator–promoter-core and repressor–operator pairs that determine transcriptional repression. The [Activator] value was constant because the transcriptional activator was expressed constitutively. A non-equilibrium term was introduced as a correction term for non-equilibrium effects. **g** and **h** Experimental measurements and parameter fitting of response functions for cI434 at $P_{T7}$ **g** and $P_{ECF11}$ **h**. *Solid lines* denote parameter fitting using a non-equilibrium correction term; *dashed* lines represent fitting without the non-equilibrium term. Data represent the means ± SD from at least three replicate experiments. NE, non-equilibrium

**Parameterizing promoter cores via biophysical modeling.** Next, we quantitatively described the process of transcription activation mediated by the interactions between the promoter cores and the corresponding $\sigma^{ECF11}$ and T7 RNAP (both referred to as transcriptional activators hereafter). An experimental framework reported in our previous study[37] was used. In this framework, the expression level of a transcriptional activator is indicated by an independently characterized circuit in which superfolder green fluorescent protein (sfGFP) is driven by the same "input promoter" (Fig. 3a). In this way, the input–output functions (called response functions hereafter) describe the intrinsic properties of interactions between the promoter cores and their cognate transcriptional activators regardless of the identity of the "input promoter".

A biophysical model that is mathematically equivalent to the conventional Hill function[10, 38] was adopted to interpret the experimentally measured response functions:

$$\text{Output} = \frac{\alpha \left( \frac{\text{Input}_A}{K_A} \right)^{n_A}}{1 + \left( \frac{\text{Input}_A}{K_A} \right)^{n_A}} + \beta \qquad (1)$$

where $\alpha$, $\beta$, $n_A$, and $K_A$ represent the maximal and basal promoter activity, the Hill coefficient, and the dissociation constant of the transcriptional activator–promoter core pair, respectively; $\text{Input}_A$ represents GFP fluorescence as a proxy for the expression level of the transcriptional activator (Fig. 3b). We found that, if $n_A$ is fixed at 1.0 (i.e. there is no cooperativity in the transcriptional

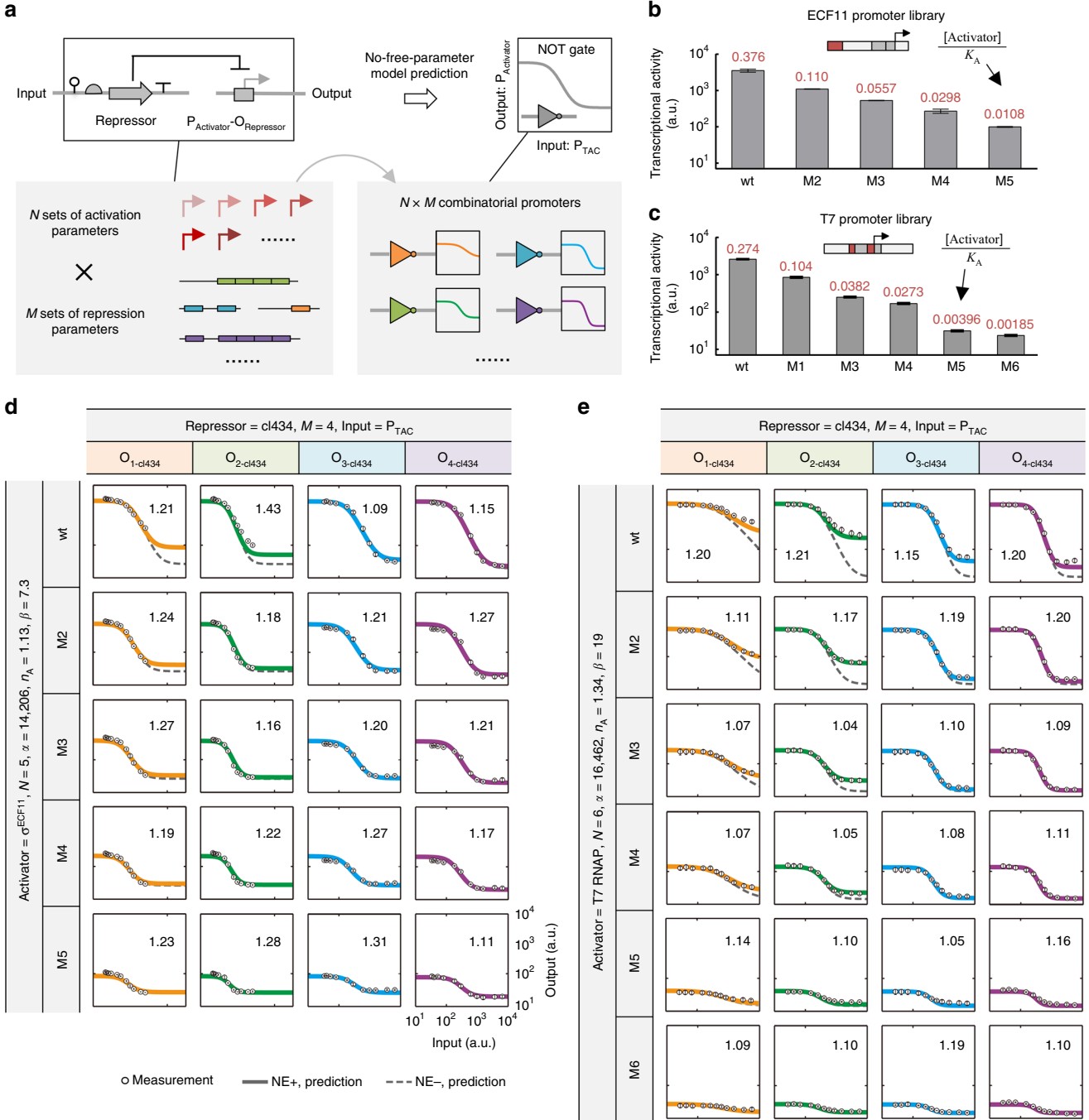

**Fig. 4** Activation- and repression-dependent parameters allow the design of combinatorial promoters. **a** Schematic representation of the combination of $N$ sets of activation-dependent parameters from promoter cores and $M$ sets of repression-dependent parameters from operators used to design $N \times M$ combinatorial promoters encoding NOT-gate functions. **b** and **c** Construction, characterization, and parameterization of a $P_{ECF11}$ **b** and a $P_{T7}$ promoter library **c**. The promoter region subjected to mutagenesis is highlighted in *red*. The corresponding value for [Activator]/$K_A$ is provided for each mutant. **d** Experimentally determined and predicted response functions of combinations of five $\sigma^{ECF11}$ promoter cores and four cl434 operators. Mean relative (-fold) error $= 10^{\frac{1}{Q}\sum|\log_{10}\text{Measurement}-\log_{10}\text{Prediction}|}$, whereby $Q$ represents the number of data points for each response function. **e** Experimentally determined and predicted response functions of combinations of six T7-RNAP promoter cores and four cl434 operators. The mean relative($n$-fold) error for the prediction of each response function is given. Data represent the means $\pm$ SD from at least three replicate experiments. NE+, with non-equilibrium correction term; NE−, without non-equilibrium term

activation), the best-fit parameters sufficiently interpreted the experimental data for $\sigma^{ECF11}$ (Fig. 3c). However, a model with such a value of $n_A$ could not explain the slight but discernible sigmoid feature in the response function of T7 RNAP (Fig. 3d). Mathematically, this issue cannot be resolved unless $n_A$ is relaxed for parameter fitting. We found that the best-fit value of $n_A$ is

1.34, suggesting that there is weak cooperativity in transcription activation by T7 RNAP (Fig. 3d). To assess whether the cooperativity is a result of cell growth retardation[39] in response to the burden/toxicity of T7 RNAP, we calculated the relative growth rates of the samples described in Fig. 3c, d and plotted them against the expression level of T7 RNAP (Supplementary

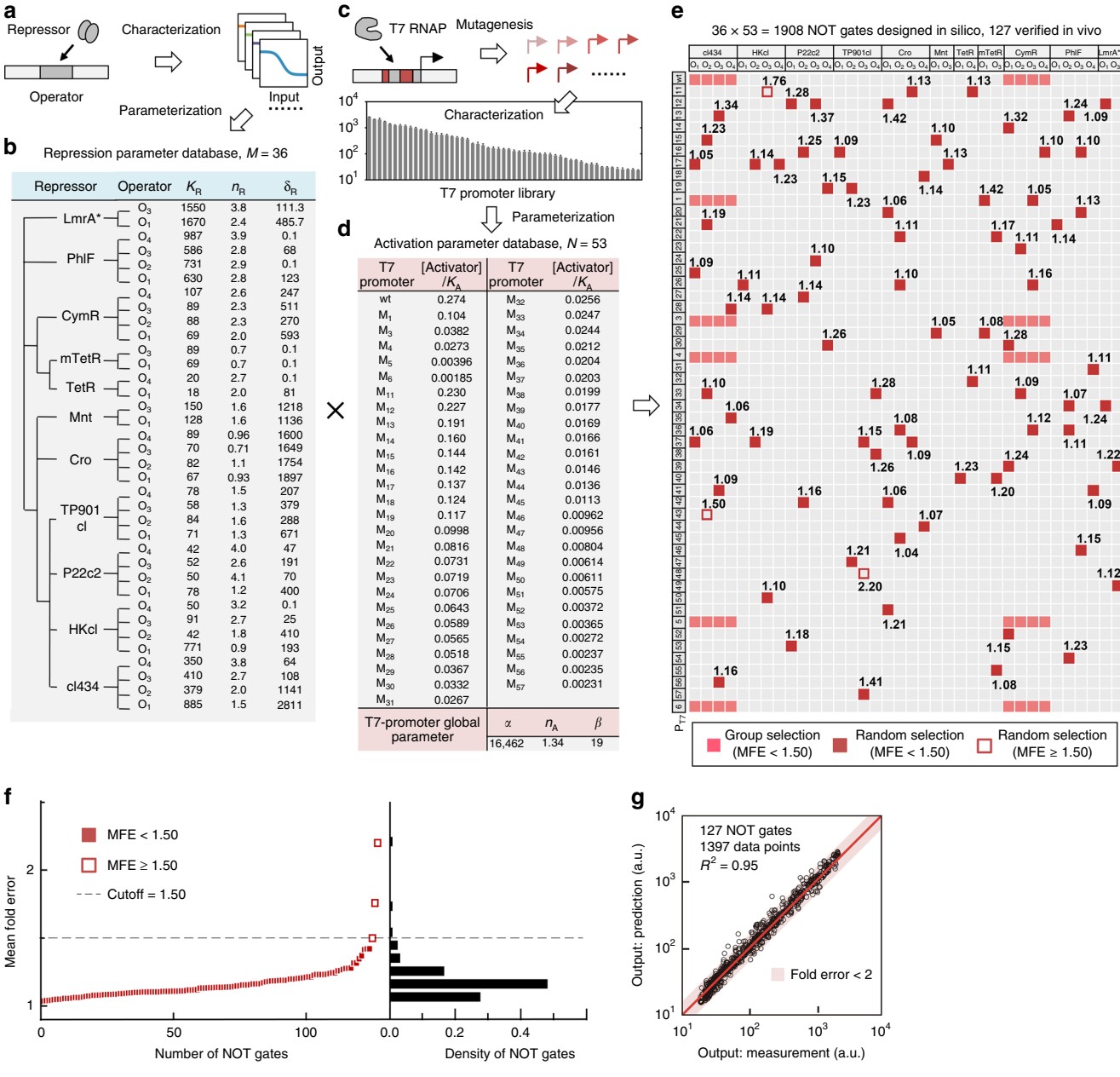

**Fig. 5** High-precision manufacturing of combinatorial promoters encoding NOT-gate functions. **a** Flow diagram depicting creation of the repression-dependent parameter database. **b** Repression database consisting of 36 sets of repression-dependent parameters with diverse dynamic features. The parameter values are given for each repressor–operator pair. **c** Flow diagram illustrating construction of the repression-dependent parameter database. **d** Activation database consisting of 53 sets of activation-dependent parameters. The value for [Activator]/$K_A$ is provided for each T7-RNAP promoter core; the values of $\alpha$, $n_A$ and $\beta$ are constant throughout the library. **e** Combination of activation- and repression-dependent parameter databases to design combinatorial promoters that function as NOT gates. The combinations selected for experimental evaluation are highlighted in *pink* and *red*. *Pink rectangles* denote combinations selected as groups (data for cl 434 operators are shown in Fig. 4e; data for cymR operators are shown in Supplementary Fig. 5). *Red rectangles* or *boxes* denote randomly selected combinations. The response functions were measured, and the corresponding mean relative(-fold) error (MFE) values were calculated for each selected combinational promoter. **f** Illustration of the achieved precision of promoter design for all 127 combinatorial promoters. A total of 124 of the promoter designs exhibited mean relative(-fold) errors <1.5, with most having error values between 1.1 and 1.2. **g** Comparison of experimental measurements with model predictions for all 1397 data points for 127 combinatorial promoters encoding NOT-gate functions. $R^2$ was calculated by linear regression. At least three biological replicates were used to measure each promoter's response function. The data shown in **b** and **d** were obtained from model fitting of experimental measurements. The MEF in **e** and **f** was calculated as described in the legend for Fig. 4

Fig. 3). The results disproved this hypothesis. In addition, re-examining the parameter fitting for $\sigma^{ECF11}$ revealed that it is insensitive to the relaxation of $n_A$. Based on these observations, $n_A$ was relaxed for both $\sigma^{ECF11}$ and T7 RNAP in the subsequent parameter fitting.

It is notable that the abovementioned four parameters ($\alpha$, $\beta$, $n_A$, and $K_A$) are sufficient to describe the interactions of ECF11 and T7 RNAP with their respective promoter cores. Therefore, the biological roles of promoter cores in transcription initiation can be parameterized as a table of "transcriptional activation-

dependent parameters" (Supplementary Table 3) for which the values would be constant regardless of the operators they are to be combined with.

**Parameterizing operators via biophysical modeling**. For the characterization of operator–repressor interactions that mediate transcriptional repression, an experimental framework similar to that for transcriptional activation was used (Fig. 3e), with the following differences: (i) the expression of repressors such as cI434 was varied; (ii) the "output promoter" was the combination of an operator and a given promoter core; and (iii) $\sigma^{ECF11}$ and T7 RNAP were constitutively expressed.

According to the promoter architectures that we have designed (Fig. 2g), repressors would inhibit transcription by simply binding to their cognate operators, thus restricting the access of activators to the promoter core (Fig. 3f). Due to this simplicity, the transcriptional repression can be described by introducing new parameters into Eq. (1):

$$\text{Output} = \frac{\alpha \left(\frac{[\text{Activator}]}{K_A}\right)^{n_A}}{1 + \left(\frac{[\text{Activator}]}{K_A}\right)^{n_A} + \left(\frac{\text{Input}_R}{K_R}\right)^{n_R}} + \beta \qquad (2)$$

where [Activator] is a constant due to the constitutive expression of a transcriptional activator; and $K_R$ and $n_R$ represent the dissociation constant and Hill coefficient of a repressor's binding to its cognate operator, respectively. In Eq. (2), $\alpha$, $\beta$, $n_A$, and $K_A$ are the same as described for Eq. (1), whereas $K_R$ and $n_R$ describe the interactions between a repressor and its cognate operator that mediate transcriptional repression (Fig. 3f). We substituted the values of the activation-dependent parameters into Eq. (1) to solve [Activator] and further into Eq. (2), seeking to fit repression-dependent parameters. However, we found that Eq. (2) could not satisfactorily fit the response functions of repressors such as cI434 when the transcriptional activator was T7 RNAP (Fig. 3g). This inconsistency derives from the operator-dependence of basal transcriptional activity at saturated levels of repressor expression, because the Output values are supposed to converge at the value of $\beta$ when $\text{Input}_R$ is sufficiently high (Fig. 3g, *gray dashed lines*). This suggests that, in addition to the conventional equilibrium assumption, possible non-equilibrium competition between transcriptional activators and repressors should also be taken into consideration. For instance, during the DNA replication process, most DNA-bound proteins temporally detach from the DNA, which creates a time window (usually called relaxation time) for both transcriptional activators and repressors to re-bind to the promoter[40]. Therefore, we introduced a correction term into Eq. (2) to describe such non-equilibrium effects:

$$\text{Output} = \frac{\alpha \left(\frac{[\text{Activator}]}{K_A}\right)^{n_A}}{1 + \left(\frac{[\text{Activator}]}{K_A}\right)^{n_A} + \left(\frac{\text{Input}_R}{K_R}\right)^{n_R}} + \beta + \left(\frac{\text{Activator}}{K_A}\right) \cdot \delta_R \tag{3}$$

where $\delta_R$ can be interpreted as the relaxation time of a repressor re-binding to its operators. As shown in Fig. 3f, the first two terms represent an equilibrium-state description of transcriptional regulation and basal transcriptional activity, respectively; the third term is the newly introduced correction term for the non-equilibrium effect.

When this refined model was applied to the operators of cI434 with T7 RNAP as the activator, precise fitting was easily achieved for all four response functions (Fig. 3g, *solid lines*). When $\sigma^{ECF11}$ was used as the activator and $P_{ECF11}$ as the promoter core,

the response function fitting for cI434 operators was insensitive to the addition of the non-equilibrium term: fitting using Eq. (3) was satisfactory, whereas fitting with Eq. (2) resulted in only merely a slight decrease in precision (Fig. 3h). These results confirmed that our new model for transcriptional repression is reliable for both ECF11 and T7 RNAP. As such, the values of repression-dependent parameters that recapitulate the interactions of transcriptional repressors with their cognate operators can also be obtained via model fitting without introducing free parameters (Supplementary Table 3).

We also measured the response functions of cI434 operators using T3 and gh-1 RNAPs as the transcriptional activators and applied Eq. (3) to fit them. Interestingly, in both cases, the response functions could be satisfactorily fit using the new rather than old model (Supplementary Fig. 4), suggesting that the non-equilibrium effect is a common transcriptional activation feature of T7-family RNAPs.

**Prediction for promoter core and operator combinations**. In the non-equilibrium term of our model, the activation- and repression-dependent parameters are interconnected by $\delta_R$ (Fig. 3f). For a given transcriptional activator, however, $\delta_R$ is solely repression-dependent. Therefore, we hypothesized that, for a given transcriptional activator (e.g. $\sigma^{ECF11}$ or T7 RNAP), $N$ sets of activation-dependent parameters from promoter cores and $M$ sets of repression-dependent parameters from operators could be combined to design $N \times M$ novel combinatorial promoters with parameter-free prediction power (Fig. 4a). To test this hypothesis, five promoter core mutants for $\sigma^{ECF11}$ and six for the T7 RNAP with 38- and 108-fold variation in transcriptional activity, respectively, were characterized and parameterized using the abovementioned experimental and modeling framework (Fig. 4b, c). Within each group, the values of $\alpha$, $\beta$, $n_A$, and [Activator] were identical; the only difference was the value of $K_A$ (Fig. 4b, c). When integrating the promoter cores of $\sigma^{ECF11}$ with the operators of cI434 to design combinatorial promoters encoding NOT-gate functions, we were surprised by the extraordinary precision of model prediction regarding the promoter response curves (Fig. 4d). Afterwards, the mean fold error of prediction against the measurements was calculated for every response curve and we found that it was ≤1.43 throughout all 20 combinatorial promoters (Fig. 4d). The precision of model prediction for promoter design using T7 RNAP-dependent promoter cores was similar, or even higher, with mean errors ≤1.21-fold throughout all 24 NOT-gate promoters (Fig. 4e). The ability to make precise predictions about these combinatorial promoters demonstrates that, for a given transcriptional activator, the activator- and repressor-derived parameters together are sufficient to provide a complete picture of the outcome of their combination.

The necessity of the non-equilibrium correction term containing "$\delta_R$" in our model was also highlighted. If the promoter response curves were calculated without including this term (Eq. (2)), predictions about the combinatorial promoters using $\sigma^{ECF11}$-dependent promoter cores roughly matched the measurements (Fig. 4d, *dashed lines*); for those using T7 RNAP-dependent promoter cores, however, the resulting predictions were far from precise (Fig. 4e, *dashed lines*), indicating an unambiguous requirement for the non-equilibrium term in T7 RNAP-dependent transcriptional activation.

**Designing genetic NOT gates at sub-promoter resolution**. Putting the precision of promoter design at sub-promoter resolution into the context of genetic circuit engineering, we realized that we might be able to transform the process of circuit engineering from the conventional design-build-test-learn cycle to a

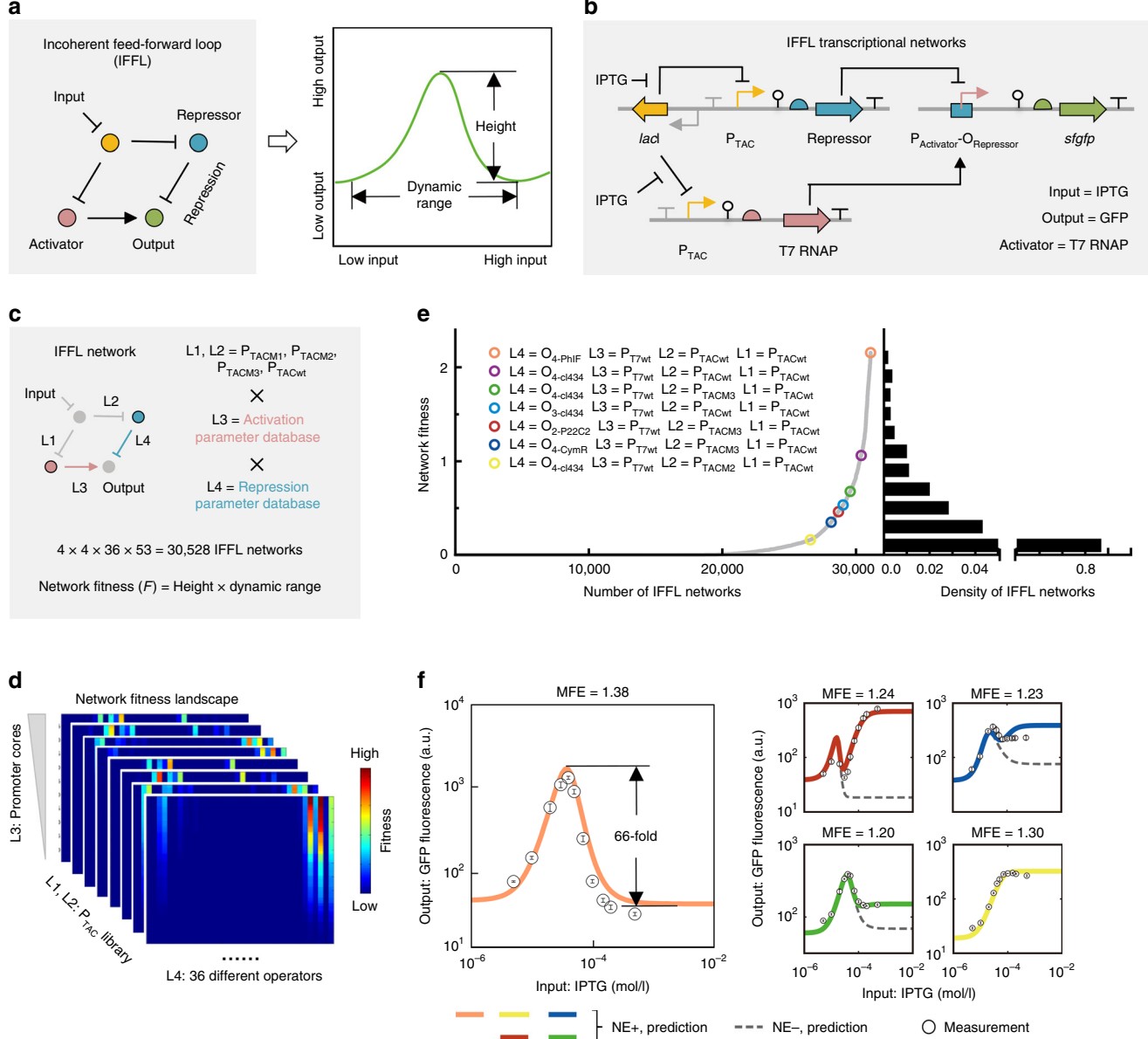

**Fig. 6** Precise design of transcriptional IFFL networks. **a** Schematic of the utilized four-node network with incoherent-feedforward-loop (IFFL) topology and the predicted network response function. **b** Genetic implementation of transcriptional IFFL networks. **c** Schematic representation of the network link assignment, design space, and computation task. The parameters required for the computation tasks were retrieved from the databases and substituted into the transcriptional IFFL network model to traverse the entire design space. **d** Computationally obtained network fitness landscape encompassing all 30,528 IFFL designs. **e** Rank and distribution of network fitness for all 30,528 IFFL designs. The seven network designs, selected for biological implementation, are indicated with *open circles. Purple* and *cyan circles* denote the two designs shown in Supplementary Fig. 9; other designs selected to have different levels of network fitness are shown in **f**. **f** Experimental and predicted response functions of the IFFL network design with the greatest stripe-forming ability and four other designs used as controls. The color code is the same as in **e**; the design shown in salmon was predicted to be the best performer. Data represent the means ± SD from three replicate experiments conducted on different days. MFE, mean relative(-fold) error; NE+, with non-equilibrium correction term; NE−, without non-equilibrium term

much simpler, mix-and-match workflow. As a proof of concept, we sought to manufacture combinatorial promoters encoding genetic NOT-gate functions using a large quantity of insulated transcriptional elements (Fig. 5). The table of repression-dependent parameters was expanded to include all 11 repressors and their 36 corresponding synthetic operators via the abovementioned experimental and mathematical methods (Fig. 5a); this yielded a parameter database for designing transcriptional repression with considerable diversity of dynamic features (Fig. 5b; see results for individual repressor–operator pairs in Supplementary Fig. 5). With respect to transcriptional

activation, T7 RNAP was selected as the activator and the corresponding promoter cores were taken from a library of 53 T7 promoter mutants (Fig. 5c). A database of activation-dependent parameters, with a >100-fold variation in the value of the [Activator]/$K_A$ term, was computationally abstracted from the characterization data of this library (Fig. 5d; see raw data in Supplementary Fig. 5); the promoter core ($P_{T7M1}$) used to obtain the repression-dependent parameter database was not included.

Next, we computationally enumerated all 1908 possible promoter designs, randomly selected 83 of these for biological implementation, experimentally characterized their NOT-gate

functions, and compared the measurements with model predictions (Fig. 5e; see results for individual promoters in Supplementary Fig. 6). The results showed that, despite the wide parameter ranges of these promoters (Supplementary Table 5), 80 were precisely predicted to function with mean errors <1.50-fold, corresponding to a success rate of >96% (Fig. 5e, f). It should be noted that the three exceptions ($P_{T7M43}$ + $O_{2\text{-}cI434}$, $P_{T7M11}$ + $O_{3\text{-}HKcI}$, and $P_{T7M48}$ + $O_{3\text{-}TP901cI}$) had 1.50-, 1.76-, and 2.20-fold mean errors of model prediction, respectively, which means that they were also firmly within the same order of magnitude, albeit beyond the cutoff. To illustrate the design precision more intuitively, the predictions and measurements for all 127 T7 RNAP-dependent NOT gates, including those shown in Fig. 4e, were subjected to a linear regression analysis. Notably, the $R^2$ value was 0.95 across the entire set of 1397 data points (Fig. 5g).

**Designing transcriptional networks with complex functions.** We next asked whether the insulated transcription elements and no-free-parameter modeling framework could be used to precisely design complex transcriptional networks. As a proof of concept, we attempted to design four-node transcriptional networks with IFFL network topology. In these networks, only the intermediate (and not the high or low) level of input can activate the output (Fig. 6a). Such a function is designated "stripe forming" because the natural counterpart networks involved in metazoan embryonic development often interpret a morphogen concentration gradient as a central single stripe of gene expression[41–43].

To test whether our no-free-parameter modeling framework still precisely described transcriptional IFFL networks, the four nodes were assigned as follows: lacI to receive an IPTG input, cI434 as the repressor, T7 RNAP as the activator, and sfGFP as the output (Fig. 6b). Considering the high efficiency of T7 RNAP in transcription initiation, two operators, $O_{3\text{-}cI434}$ and $O_{4\text{-}cI434}$, were selected in order to achieve the strong repression needed for the stripe-forming function. Accordingly, a no-free-parameter model describing such a network designs was derived from Eq. (3). The predictive power of the model was evaluated by comparing the measurements with the predictions. The model predictions, regarding the response curves of the $O_{3\text{-}cI434}$ and $O_{4\text{-}cI434}$ IFFL networks, were both precise, with mean errors of 1.30- and 1.27-fold, respectively (Supplementary Fig. 9). It is interesting to note that if the non-equilibrium term containing "$\delta_R$" was not taken into account, complete repression of GFP expression was predicted when the expression of repressors was saturated (Supplementary Fig. 9, *dashed lines*), which obviously did not match the measurements. If, on the other hand, this term was included, the secondary increase of GFP expression in the response curves could be precisely predicted (Supplementary Fig. 9, *solid lines*), once again highlighting the necessity of this non-equilibrium term. The explanation for the secondary increase of GFP expression is straightforward: at very high IPTG concentrations, the contributions of the equilibrium terms to the overall output tends to be negligible due to the high expression level and high Hill coefficient of cI434. In such a situation, the overall output depends primarily on the contribution of $\delta_R$ (*arrows* in Supplementary Fig. 9). These results demonstrate that our modeling framework can be still highly predictive with respect to complex transcriptional networks.

Subsequently, we used our modeling framework and insulated transcriptional elements to design transcriptional IFFL networks with minimal a priori consideration, in order to determine whether the mix-and-match design process is indeed feasible. Links L1 and L2 in the IFFL networks were assigned to four

$P_{TAC}$ promoter variants with different dose–response curves (Supplementary Fig. 7). Links L3 and L4 corresponded to the transcriptional activation and repression exerted by the promoter cores and operators, respectively (Fig. 6c). The computational task was defined with the goal of identifying the transcriptional IFFL network with the greatest stripe-forming ability. Accordingly, the transcriptional activation- and repression-dependent parameters were first retrieved from the databases of T7 promoter cores and operators, together with the parameters of $P_{TAC}$ promoters, and then substituted into the model of IFFL networks in order to traverse the entire design space. The procedure yielded 30,528 different *in silico* network designs. To quantify the stripe-forming ability of these network designs, a "fitness" index was developed (Fig. 6c). The network fitness landscape revealed that the stripe-forming ability (i) strongly depends on the identity of repressor–operator pairs and (ii) is enhanced by promoter cores exhibiting strong transcriptional activity (Fig. 6d; see Supplementary Fig. 8 for full fitness landscape). Subsequently, all 30,528 network designs were ranked according to their fitness and the design with the highest fitness was implemented in vivo (Fig. 6e, $O_{4\text{-}PhlF}$ + $P_{T7wt}$ + $P_{TACwt}$ + $P_{TACwt}$). The experimental results showed that this network executes the stripe-forming function in response to IPTG concentration gradients with 66-fold pulsing (Fig. 6f). Moreover, the mean error of model prediction was as low as 1.38-fold (Fig. 6f, *left* panel). To verify the reliability of this computational search, four more network designs were selected throughout the ranking and subjected to the same experimental evaluation. As expected, both the network fitness and response functions of these network designs were precisely predicted (mean errors <1.30-fold; Fig. 6f, *right* panel).

One advantage of our insulated transcriptional element approach is that transcriptional activation and repression can be separately modulated in genetic circuits without affecting each other. We hypothesized that tuning the peak height by changing the promoter core would not affect the peak position in the IFFL circuits. This prediction was confirmed by re-design of the highest-fitness IFFL network with five new promoter cores to replace the $P_{T7wt}$ promoter (Supplementary Fig. 10). Based on the collective results, we conclude that our insulation-based, sub-promoter-resolution genetic circuit design strategy is sufficiently powerful at the level of complex networks.

**Discussion**

The engineering of promoters has thus far been a largely ad hoc exercise due to the high degree of complexity in the interactions between sub-promoter elements[22]. As a consequence, promoters are usually regarded as basic, functionally indivisible parts in conventional genetic circuit engineering, despite their multi-element nature[4, 6]. For instance, promoter design per se was not undertaken in the work of Nielsen et al.[24], even though some engineering, such as utilization of insulator sequences, was carried out to eliminate the interferences between promoters and surrounding parts. The scaling-up of genetic circuits, however, has highlighted that the limitations affecting the rational design of promoters as the information processing hubs have become technical bottlenecks and the rate-limiting step in synthetic biology[4, 44]. To address this issue, we developed an experimental and modeling framework that functionally decouples the prokaryotic promoter elements, yielding modular promoter cores and operators. These transcriptional elements, which are functionally insulated from each other, can be re-used in combination to predictably design new promoters for a variety of circuits. Following this strategy, even a small number of promoter cores and operators can be reused to substantially increase diversity in synthetic transcriptional regulation.

In synthetic biology, information about biological systems is organized using an abstraction hierarchy in order to enable researchers working at one layer of complexity to disregard the details encountered at other layers[3, 6]. Previous studies represented by Nielsen et al.[24] have developed methods for rational design from the layer of "parts" to the layer of "circuitry"[7]. Rational design at more basic hierarchical layers, however, remains largely unexplored. Our study achieved the functional insulation and predicable combination of "sub-parts"-level genetic components, thus pushing the resolution of genetic circuit design to a new limit. Moreover, because our method works at a different layer, it is compatible with existing genetic circuit technologies including the "Cello" and "Clotho" software[23, 24, 45].

In this work, synthetic transcriptional regulation was defined by specific interactions between defined components. For instance, transcriptional activation by T7 RNAP at the T7 promoter is fully independent of the host transcription machinery. Regarding this biological simplicity, the insulated transcriptional elements hold the promise of being transferred from E. coli to other organisms while still retaining their function. One recent study[46] showed that the performance of promoter cores, recognized by other T7-family RNAPs, is highly comparable between E.coli and phylogenetically distant bacteria ($R^2 \geq 0.94$). For ECF σs, the promoter cores function comparably when transferred from E. coli to Klebsiella oxytoca[34], probably due to the evolutionary conservation of bacterial RNAP subunits[47]. Collectively, these results provide evidence for the transferability of insulated transcriptional elements.

The insulated promoter cores used in this study were derived from T7 and ECF11 promoters. We have shown that an insulated promoter core is a common feature of T7-family RNAPs and bacterial ECF σ factors (Supplementary Fig. 2). As a result, the toolset of promoter cores can be expanded to include many other well-studied T7-family and ECF σ promoters[16, 34, 46] and further combined with "part mining" technology[48] to source more candidates for refinement, in accordance with the scale-up of circuit construction. The toolset of synthetic operators is even more extensive due to the higher abundance of available repressor–operator pairs[18, 35]. Intriguingly, regarding the simple role of operators in our promoter architectures, it is possible to incorporate dCas9-sgRNA complexes into our promoter design as the repressors to further enhance the diversity of our toolset.

We noticed that there was unusually high transcriptional activity in two failed combinatorial promoter designs ($P_{T7M43}$ + $O_{2-cI434}$ and $P_{T7M48}$ + $O_{3-TP901cI}$), even in the absence of T7 RNAP; this activity could not be explained by either the promoter core or the operator alone (Supplementary Fig. 11a). Analyses using the BPROM program revealed that, unexpectedly, $\sigma^{70}$-dependent promoters emerged at the physical interface between the promoter cores and operators (see Supplementary Table 2 for prediction results). Interestingly, when the contribution of emergent transcriptional activity was integrated into the promoter models, the precision of the predictions regarding these two promoters immediately increased (Supplementary Fig. 11b).

Unexpected cryptic promoters arising from the connection of parts were also reported in previous studies[20, 49], motivating us to re-examine all 83 combinatorial promoters in the absence of T7 RNAP (Supplementary Fig. 11c). Indeed, we found six combinatorial promoters that exhibited strong transcriptional activity, indicating that promoter emergence from the sequence composition is not a rare event and is in fact more common than one would assume based purely on the rate of significant phenotypical failures. Therefore, in large genetic circuits in which many parts are combined with various permutations, DNA sequences that resemble regulatory elements, such as promoters

and terminators, would appear at the part junctions with considerable frequency. One potential solution is to develop computational tools that scan circuit DNA sequences for such unintended regulatory elements in the design or debugging process[50]. A more straightforward solution would be to apply omics technologies, such as RNA-sequencing, to characterize and debug the genetic circuits[51–53], in order to identify any unexpected cryptic regulatory elements.

Counterintuitively, we found that the best-fitting value of $n_A$ was >1 in modeling transcriptional activation for T7 RNAP. The mechanism underpinning this phenomenon, to our knowledge, has not been elucidated. One possible explanation is that the binding of a T7 RNAP molecule to the promoter facilitates its future binding, which is supported by a previous report that exposure of T7 RNAP to the binding region of the promoter alone leads to reorganization of the T7 RNAP to facilitate its future recruitment to the promoter[54]. Another possible explanation involves DNA allostery caused by the promoter unwinding of T7 RNAP. It was recently reported that the specific binding of a protein to DNA is considerably strengthened by the deformation of the proximal double-helical structure[55]. Considering the slow rate of transition from transcriptional initiation to elongation and the high rate of promoter binding-releasing for T7 RNAP[56, 57], the relatively long-term promoter unwinding caused by the leading T7 RNAP could also facilitate the promoter binding by the subsequent T7 RNAP.

In modeling transcriptional repression, we introduced a correction term to describe the non-equilibrium effect. Accordingly, post-DNA replication competitive binding of activators and repressors was proposed as a possible explanation that proved to be consistent with our many experimental observations. For instance, operators with low repressor-binding affinity generally tend to have high $\delta_R$ values (Fig. 5b). This fits our explanation that strong repressor binding leads to a small permissive time window ($\delta$) for the activator to re-bind. Although our explanation might not be consistent with the biological scenario, the non-equilibrium term is still necessary for precise model predictions and does capture the salient features of underlying biological processes. Interestingly, the non-equilibrium effect was found to be negligible for $\sigma^{ECF11}$, indicating that our modeling is not contradictory to observations in previous studies where $\sigma^{70}$-dependent promoters were usually used, regarding the highly similar biochemical nature of $\sigma^{70}$ and ECF σ factors[58]. Differences in the non-equilibrium effect between the three types of transcriptional machinery could be important criteria for part selection in circuit design.

Regarding the future use of insulated transcriptional elements, one issue we might encounter is activator/repressor titration arising from the connection of more than one downstream promoters to a circuit[4, 59]. For example, if the repressor/activator of an IFFL network is shared by newly added output promoters, the repressor/activator would be titrated away from its original circuit; this effect is also called retroactivity[60, 61]. In such cases, the assumption that the activator/repressor copy number greatly exceeds that of the promoters, breaks down[59]. In order to retain the accuracy of predictions, the mathematical model describing the transcriptional regulation of a given promoter should be adjusted from the total concentration of the repressor/activator ([Repressor/Activator]$_{total}$) to the concentration of free repressor/activator ([Repressor/Activator]$_{free}$).

## Methods

**Strains and plasmids**. *E. coli* K-12 DH10B was used for plasmid construction, parts characterization, and circuit measurements throughout this study. This strain was also used for the construction of strains that chromosomally express phage RNAPs or ECF σ factors in a constitutive or inducible manner. The vast majority of

the plasmids used in this study were constructed using two basic vectors, pPT and pRG, which were constructed and preserved in our laboratory. pPT was derived from the standard BioBrick backbone pSB4C5 and was used to construct the cassettes for the "output promoter"-driving reporters. pRG with p15A-AmpR backbone was used as the carrier for the "input promoter" to express transcriptional repressors in response to IPTG. Golden Gate Assembly was used to incorporate the output promoter into pPT and RBS-CDS into pRG (see Supplementary Fig. 12 and Supplementary Table 4 for the detailed construction process). The collection of plasmids and host strains, as well as details of their use in each experiment, are described in "Detailed plasmid specifications" below.

**Detailed plasmid specifications**. For identification of insulated promoter cores and the refinement of synthetic operators, the constitutively expressed RNAP was integrated into the chromosome of *E. coli* DH10B using pOSIP (for T7RNAP and $\sigma^{ECF11}$) or cloned into the plasmid pRG (for T3, gh-1, MmP1, $\sigma^{ECF16}$, and $\sigma^{ECF20}$). Saturation mutagenesis of a given promoter segment was conducted using degenerate primers via PCR annealing. Specifically, a random DNA sequence corresponding to the segment was designed near the 5' end of one primer, and the sequence of the other primer was defined; these two primers were annealed at their 3' ends and further extended using Phusion DNA polymerase (NEB, M0530) to create the entire promoter region. The resulting promoter mutation library as a mixture was then cloned into the plasmid pPT via Golden Gate Assembly for the subsequent colony picking and in vivo analysis.

For characterization of transcriptional activation, the $P_{TAC}$-RNAP cassette was integrated into the chromosome of *E. coli* DH10B using pOSIP, and the plasmid carrying the pPT-promoter core cassette was introduced into the resulting strain, along with the plasmid pRGc (pRG plasmid with the $P_{TAC}$-*lacZ*α fragment deleted).

For characterization of transcriptional repression, the *RNAP* genes were all integrated into the chromosome of *E. coli* DH10B using pOSIP for constitutive expression, and the pPT-derived plasmids carrying the combinational promoters were introduced into the resulting strain, along with the pRG-derived plasmid carrying the corresponding repressor.

For construction of dose–response curves for the "input promoters", the RNAP/ repressor gene downstream of the $P_{TAC}$ promoters was exchanged for *sfgfp*. A blank plasmid (pPTc), containing the same replication origin and resistance gene as pPT, was co-transformed with the "input promoter" plasmid into the corresponding strains to standardize the antibiotics usage.

The plasmid specifications of the transcriptional IFFL networks were similar to those used for the characterization of transcriptional repression, except that the cassette integrated into the chromosome was the same as that used for the characterization of transcriptional activation. The plasmids used in each experiment discussed above are summarized in Supplementary Table 5. The sequences of important cassettes are summarized in Supplementary Table 6 and Supplementary Table 7.

**Cell growth and fluorescence measurement**. All incubations were carried out using a Digital Thermostatic Shaker (AOSHENG) maintained at 37 °C and 1000 r. p.m., using Corning flat-bottom 96-well plates sealed with sealing film (Corning, BF-400-S). For characterization of parts and circuit response functions, a previously developed quantitative method[37] that measures gene expression at steady state was used. Briefly, bacteria harboring the parts or circuits of interest were first inoculated from single colonies into a flat-bottom 96-well plate for overnight growth, after which the cell cultures were diluted 196-fold with M9 medium. After 3 h of growth, the cultures were further diluted 700-fold with M9 medium, and incubated for another 6 h. Finally, 20-μl samples of each culture were transferred to a new plate containing 180 μl per well of PBS supplemented with 2 mg ml⁻¹ kanamycin to terminate protein expression. The fluorescence distribution of each sample was assayed using an LSRII flow cytometer (BD Biosciences) with appropriate voltage settings; each distribution contained >20,000 events. Each sample was experimentally assayed at least three times. The arithmetical mean of each sample was determined using FlowJo software (v7.6). For identification of insulated promoter cores and operator refinement, the quantitative steady-state method was adjusted to allow for high-throughput measurement. The differences were: (i) the culture duration after the second dilution was decreased to 4.5 h; and (ii) cell fluorescence was assayed using a Synergy Mx microplate reader (BioTek) and normalized to the $OD_{600}$ (normally 0.4–0.6) for each sample.

**Media and buffers**. All chemicals used in the study were purchased from Sigma-Aldrich unless stated otherwise. LB medium: 10 g l⁻¹ tryptone, 5 g l⁻¹ yeast extract, and 10 g l⁻¹ NaCl. For agar plates, 15 g l⁻¹ agar was added. M9 medium: 6.8 g l⁻¹ $Na_2HPO_4$, 3 g l⁻¹ $KH_2PO_4$, 0.5 g l⁻¹ NaCl, 1 g l⁻¹ NH4Cl), 0.34 g l⁻¹ thiamine, 0.2% casamino acids (BD Biosciences), 0.4% glucose, 2 mM $MgSO_4$, and 100 μM $CaCl_2$. Antibiotic concentrations: where appropriate, the media contained ampicillin at a final concentration of 100 μg ml⁻¹ from a 100 mg ml⁻¹ aqueous stock solution and chloramphenicol at a final concentration of 25 μg ml⁻¹ from a 34 mg ml⁻¹ stock solution in absolute ethanol; for pOSIP-KO-mediated chromosomal integration, 50 μg mL⁻¹ kanamycin-sulfate was used. The indicated concentrations of antibiotics were used for both LB and M9 media. Inducer concentration: an isopropyl-

d-1-thiogalactopyranoside gradient was prepared at 0, 1, 5, 10, 20, 30, 50, 70, 100, 200, and 500 μM final concentrations, diluted from a stock in DMSO. Phosphate-buffered saline (PBS): 8 g l⁻¹ NaCl, 0.2 g l⁻¹ KCl, 1.44 g l⁻¹ $Na_2HPO_4$, and 0.24 g l⁻¹ $KH_2PO_4$. Kanamycin (2 mg ml⁻¹) was added to the PBS before sampling in order to terminate protein expression.

**Data analysis and modeling**. The parameters in Eqs (1–3), describing the dynamic features of promoter cores and operators, were obtained by fitting using the "fminsearch" function in MATLAB (version 2010a). The predictions for the combinations of promoter cores and operators were performed using the same function, except that the parameter values were retrieved from the fitting results of characterization data. Modeling of the four $P_{TAC}$ promoter variants was adapted from the Supplementary Eq. (7) of a previous study[21], with the following modification: the value of one parameter, $N_D$, was fixed at 1. The steady-state solutions for transcriptional IFFL networks were calculated using MATLAB (version 2010a). The fitness function was calculated by directly multiplying the widths and heights of the peaks in the response functions, wherein the width was defined as the magnitude of the input change when the output began to increase from 0.1-fold of the maximum value, and subsequently reverted to this value, and the height was defined as the magnitude of the output change when the output increased from the minimum to the maximum value (left height) or decreased from the maximum to the minimum value (right height). Therefore, the mathematical expression of the fitness function is width×(left height)×(right height).

**Data availability**. We declare that all relevant data supporting the findings of this study are available within the article and its Supplementary Information Files or from the corresponding authors upon request.

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

## Acknowledgements

This study was supported by the NSFC (No. 11074009 to Q.O., No. 11434001 to Q.O., and No. 31470818 to Chunbo L.), the "863" Program (No. 2012AA02A702 to Q.O.), MSTC (No. 2011CBA00805 to Q.O., Nos. 2013CB734001 and 2015CB910300 to Chunbo L.), and the CAS Interdisciplinary Innovation Team (No. Y429012CX8 to Chunbo L.).

## Author contributions

Chunbo L., H.M.Z., and Q.O.: Conceived and supervised the project. Y.Z., H.M.Z., X.J., J. H., and X.G.: Designed and performed the experiments. C.L. and Chunbo L.: Constructed the biophysical model. H.M.Z., Y.Z., and Chunbo L.: Wrote the manuscript.

## Additional information

**Competing interests:** The authors declare no competing financial interests.

