## [Peer Review File · Nature Communications]

Reviewers' Comments:

Reviewer #1 (Remarks to the Author)

This manuscript describes the design of synthetic circuits in *E. coli* through the model-based performance design of synthetic parts (including insulated promoters). While the manuscript would be of some interest in the field, the current form is rather dense and hard to follow. Specifically, much of the text is rather vague and had to be read several times to gain meaning.

More importantly, comparisons with state of the art are missing. In particular, the authors cite reference 24 that also looks at a predictive design of synthetic circuits. How does the approach that the authors present here compare to this approach. The NOT gates and IFFL systems that are being designed in this paper should also be designed by the CLOTHO approach and predictive power should be compared.

The authors employ an interesting "sensitivity analysis" of key base pairs in promoters to see interactions. However, testing was not done to see if these "insulated" elements also behave the same way in different plasmid context and in the genome. Such work would need to be conducted.

The data in Figure 6 is rather confusing. How many of these IFFL were actually tested and experimentally validated? The >30K number provided in the abstract is not exactly accurate for the amount explored and seems misleading.

A hill coefficient model was used to describe the promoters, but the authors find a hill coefficient of 1.3 (no error was provided in this fit) which suggested limited interaction. Could this mean that the mechanism inherent in a Hill model is not accurate in describing the phenomena here? A more biophysical model and understanding of the promoters should be considered here.

Reviewer #2 (Remarks to the Author)

The manuscript by Zong, Zhang et al. is a well-written and very well presented description of important work in synthetic biology, particularly for future applications of synthetic biology in *E. coli* bacteria. The authors set out to solve the long-standing challenge of building regulated promoters from first principles in a manner that allows them to predictably mix and match the activator and repressors of the promoter and in doing so tune the expression output and on/off characteristics as desired. Importantly they also combine this excellent work with a mathematical model which fits the subsequent *in vivo* data with "extraordinary" precision. This work yields some new insights into how promoters work in *E. coli* and generates new parts that will undoubtedly be used by others in the field. To demonstrate a use, the authors include one of their new promoters in the construction of a feed-forward loop gene circuit and illustrate how the ability to model and predict promoter performance ahead of construction can allow selection of parts that will give the desired phenotype.

Overall, this is very high quality synthetic biology research that will be of interest to many in the field. However, I am concerned that the way it is presented makes it is quite specialised and maybe better suited to a journal specifically for synthetic biology research. The authors may want to revise the manuscript in a way that considers what the broader impact of their work is beyond bacterial synthetic biology. Is the impact the novel insights achieved by matching the modelling and data so well? How well would this work translate to promoters from other organisms? What happens when multiple instances of their designed promoters are used together to build a genetic

circuit (rather than just one as in the Figure 6 circuit)? What can now be achieved with these promoters that was not possible before?

As well as this overall major point, I have a few minor points as well – with points 6 and 7 being particularly important.

1. Line 103 – there should be a space between numbers and units – e.g. 18 bp, or 10 mM. This needs to be fixed throughout the manuscript and supplementary information.

2. Line 113 – a word is missing here. Possibly 'T7'

3. Line 127 – please add more description as to how saturation mutagenesis was done

4. Line 165 – the information in the sentence beginning on this line should be put towards the beginning of the section and not within it.

5. Lines 174 to 178 – this text seems a bit like it has been taken from a PhD thesis and isn't part of the paper. It refers to 'chapters' for example.

6. Line 200 – The slight cooperative effect with T7 polymerase may be explained by the negative effect it has at high-levels on the growth of E.coli. If T7 polymerase is expressed at high levels or if it activates high-levels of GFP expression from a strong version of a T7 promoter, it can burden the cells, slow the growth rate of the E.coli down and as a result, GFP per cell increases because the cell division rate lowers (lowering the GFP dilution rate). This is a published phenomenon noticed first by Lingchong You's group: Tan, C., Marguet, P. & You, L. Emergent bistability by a growth-modulating positive feedback circuit. *Nat Chem Biol* 5, 842-848 (2009).

7. Considering the above point, I would like to also be assured that the various activators/repressors and operator/promoters constructed in this manuscript don't negatively affect the growth rate of E.coli either through burden or off-target effects. The way the data are presented here in the paper makes it impossible to determine if there are any changes to E.coli growth when the promoters are used. Changes to E.coli growth rates will have important downstream consequences such as altering the dilution rate of proteins in gene networks. It is well established that T7 polymerase is challenging for E.coli cells due to driving extraordinarily high gene expression levels and because it transcribes RNA at a rate approximately 3 times faster than normal E.coli RNA polymerase, causing significant stress to the cell and decoupling transcription from translation.

8. Line 260 – is the model also accurate for the other activators? (the ones derived from other ECFs and T7 polymerase)

9. Line 277 – what are the black arrows pointing to in Figure 4B and 4C. Are they necessary?

10. Line 302 – "one-step to reach" – I've never heard of this phrase and I don't think it can be justified as there are more than one steps involved. Considering an alternative phrase would be wise.

11. Line 328 – in this section (everything for Figure 6) why choose to build a genetic circuit that only uses 1 of the promoters? Building one with several of the different new designed promoters would've been better. The work in this section, while very good is actually much more dependent on the performance characteristics of the pTAC promoters, rather than those developed in this study.

12. Line 419 – please insert a relevant citation for "part-mining"

13. Line 444 – another solution would be to characterise circuits/promoters using RNAseq which would identify unexpected cryptic promoters.

Point-by-point responses to the reviewers.

Questions/comments from Reviewer #1 (related comments may be grouped together):

Question 1:

This manuscript describes the design of synthetic circuits in E. coli through the model-based performance design of synthetic parts (including insulated promoters). While the manuscript would be of some interest in the field, the current form is rather dense and hard to follow. Specifically, much of the text is rather vague and had to be read several times to gain meaning.

Response:

We thank the reviewer for this helpful comment. We have revised the wording and expression throughout the manuscript, in order to make the text more comprehensible. In particular, technical jargon has been avoided as much as possible and clarified if its usage is necessary. Besides, we have used professional language editing service on our manuscript to further improve the language quality.

Question 2:

More importantly, comparisons with state of the art are missing. In particular, the authors cite reference 24 that also looks at a predictive design of synthetic circuits. How does the approach that the authors present here compare to this approach. The NOT gates and IFFL systems that are being designed in this paper should also be designed by the CLOTHO approach and predictive power should be compared.

Response:

We thank the reviewer for this critical comment which led us to clarify in the revised manuscript what the difference we made. Reference 24 and our work are not competitors because these two studies focused on different issues. Reference 24 (A.A.K. Nielsen et. al., 2016) focused on the integration of readily available biological parts (including regulatable promoters, insulators and terminators) and design principles/constraints to build genetic circuits with minimal human supervision. In reference 24, the regulatable promoters were regarded as basic, functionally indivisible parts, despite that they were actually multi-component. In our work, we focused on decomposing these regulated promoters whose components are strongly coupled and hard to dissect in function. We developed an experimental and modeling framework to decouple the components of regulatable promoters, yielding promoter cores (responsible for transcriptional activation) and operators (responsible for transcriptional repression). These transcriptional elements that are functionally insulated from each other were further re-used in combination to predictably design new promoters. Therefore, we could say that our work provides a method (to design regulatable promoters) that is backward compatible with previously reported circuit design methods including the “Cello” reported in ref. 24 and the “Clotho”.

Moreover, the design of NOT gates and IFFL networks in our study was conducted merely in order to verify the reliability of our insulated transcriptional elements, rather than to automate the genetic circuit design. To emphasize our insulation strategy, the manuscript title that was previously misleading has been changed to “Insulated transcriptional elements enable precise design of genetic circuits”. We have also re-written the first two paragraphs of Discussion, in order to thoroughly discuss the difference and the connection between the achievements of previous studies represented by ref. 24 and that of ours.

Question 3:

The authors employ an interesting “sensitivity analysis” of key base pairs in promoters to see interactions. However, testing was not done to see if these “insulated” elements also behave the same way in different plasmid context and in the genome. Such work would need to be conducted.

Response:

We appreciate the reviewer’s suggestion, and have conducted segment-by-segment saturation mutagenesis of P_{ECF11} on a new plasmid backbone (p15A origin, medium-copy number). The new results are basically the same as those in Fig. 2d (pSC101 origin, low-copy number), as shown below:

These results have been included in Supplementary Information (Supplementary Fig. 1) and the corresponding discussion has been added to the manuscript text.

Question 4:

The data in Figure 6 is rather confusing. How many of these IFFL were actually tested and experimentally validated? The >30K number provided in the abstract is not exactly accurate for the amount explored and seems misleading.

Response:

We have removed this misleading number from the Abstract. In the primary submission, we experimentally tested seven IFFL network designs as indicated in Fig. 6e; of these, two were shown in Supplementary Fig. 9 and the other five were shown in Fig. 6f. For the revision of this manuscript, we have experimentally tested five more different network designs in response to Q.16 of Reviewer #2. The newly obtained results have been included into the Supplementary Fig. 10.

Question 5:

A hill coefficient model was used to describe the promoters, but the authors find a hill coefficient of 1.3 (no error was provided in this fit) which suggested limited interaction. Could this mean that the mechanism inherent in a Hill model is not accurate in describing the phenomena here? A more biophysical model and understanding of the promoters should be considered here.

Response:

We have calculated the fitting error of n_A . The exact value of n_A is 1.34 ± 0.043 with 95% confidence bounds, which implies that there is indeed weak cooperativity in transcription initiation at the T7 promoter by T7 RNAP. This does not mean that the Hill function is inaccurate in describing T7 RNAP-dependent transcriptional

activation; instead, it means the inaccuracy of generally assumed biochemical picture that T7 RNAP acts as independent monomers in transcription initiation ($n_A = 1.0$). Some direct or indirect interactions between T7 RNAP molecules must happen; Hill function with $n_A = 1.3$ is a kind of phenomenological description but it does capture the dynamic features of underlying biological process. The mechanism underpinning this phenomenon ($n_A = 1.3$), to our knowledge, is not clear. One possible explanation is that the binding of a T7 RNAP molecule to the promoter would facilitate its future binding, because it was previously reported that exposure of T7 RNAP to the binding region of promoter alone can reorganize the T7 RNAP to facilitate its future recruitment to promoter¹. Another possible explanation is DNA allostery caused by the promoter unwinding of T7 RNAP. It was recently reported that the specific binding of a protein to DNA can be considerably strengthened by the deformation of double-helical structure nearby². Considering the slow rate of transition from transcriptional initiation to elongation and the fast rate of promoter binding-unbinding for T7 RNAP^{3,4}, the relatively long-term promoter unwinding caused by a leading T7 RNAP may facilitate the promoter binding of the following. We have re-written the first paragraph of Discussion to include these discussions into the revised manuscript.

Questions/comments from Reviewer #2 (related comments may be grouped together):

Question 1:

The manuscript by Zong, Zhang et al. is a well-written and very well presented description of important work in synthetic biology, particularly for future applications of synthetic biology in E. coli bacteria. The authors set out to solve the long-standing challenge of building regulated promoters from first principles in a manner that allows them to predictably mix and match the activator and repressors of the promoter and in doing so tune the expression output and on/off characteristics as desired. Importantly they also combine this excellent work with a mathematical model which fits the subsequent in vivo data with “extraordinary” precision. This work yields some new insights into how promoters work in E. coli and generates new parts that will undoubtedly be used by others in the field. To demonstrate a use, the authors include one of their new promoters in the construction of a feed-forward loop gene circuit and illustrate how the ability to model and predict promoter performance ahead of construction can allow selection of parts that will give the desired phenotype.

Overall, this is very high quality synthetic biology research that will be of interest to many in the field. However, I am concerned that the way it is presented makes it quite specialised and maybe better suited to a journal specifically for synthetic biology research. The authors may want to revise the manuscript in a way that considers what the broader impact of their work is beyond bacterial synthetic biology.

Response:

We thank the reviewer for this endorsement of the novelty and importance of our work. We have made efforts in three aspects to broaden the impact of our manuscript. First, many sentences in Introduction have been re-written, in order to clarify the necessity and the timeliness of our work. Second, wording and expression, especially the technical jargon, has been revised throughout the entire manuscript, in order to make our work more comprehensible. Third, we have re-written the Discussion section by including more discussions about the impact of our work and the

relationship between the previous studies and ours. Professional language editing service was also applied to the manuscript to improve its language quality.

Question 2:

Is the impact the novel insights achieved by matching the modelling and data so well?

Response:

The key to our success of precise promoter design is the insulation strategy. Previous studies have applied this strategy to achieve precise circuit design. For example, one of us, C. Lou with his co-workers⁵ used to develop a set of ribozyme-based insulators that functionally insulate promoters from the downstream sequence context; this enables highly predictable layering of transcriptional circuits. Another example is the work of A.A.K. Nielsen et. al.⁶, which combines genetic parts and insulation technology, thus enabling precise, automated design of genetic circuitry. In these examples, the design of genetic circuits has been already free-parameter-independent. In our work, we developed a method to decouple the components of regulatable promoters that were previously thought to be functionally “indivisible” and applied them to precisely design regulatable promoters from scratch. Our experimental and parameterizing methods used for predictable design are basically the same as previous studies⁵⁻⁷; the difference we made is pushing the insulation strategy to sub-promoter level. Therefore, we have confidence in our results. The corresponding discussions have been included in the manuscript.

Question 3:

How well would this work translate to promoters from other organisms?

Response:

This is a good question. We have experimentally transferred the library of insulated T7 promoters from *E.coli* (a gram negative bacterium) to *Streptomyces albus* J1074 (a gram positive bacterium). Results indicate that the activity of insulated promoters, accompanied by the constitutive expression of T7 RNAP, persists with a high correlation ($R^2=0.94$) across these two organisms, as shown below:

One recent study⁸ involving some of us reported that the promoters of T7-family RNAPs can be also predictably transferred from *E.coli* to *Halomonas* sp. TD01 (a halophilic bacterium) and *Pseudomonas entomophila* (a soil bacterium). All these results support the transferability of our insulated promoters. These results and discussions have been included into the revised manuscript as a new paragraph in Discussion.

Question 4:

What happens when multiple instances of their designed promoters are used together to build a genetic circuit (rather than just one as in the Figure 6 circuit)?

Response:

When multiple instances of an insulated operator/promoter core are used in a circuit, the corresponding repressor/activator would be titrated away by the newly added promoters; this effect is also called retroactivity. In such cases, the assumption that the activator/repressor is in excess of the number of binding sites would break down and the effect of activator/repressor titration is not negligible. In order to retain the accuracy of prediction, the corresponding models should be adjusted, such as by substituting the total concentration of repressor/activator with the free concentration of them. We have added the corresponding discussions to the manuscript as a new paragraph in the last section.

Question 5:

What can now be achieved with these promoters that was not possible before?

Response:

This is also a good question. In previous studies, the transcriptional activation and repression of a given promoter cannot be separately modulated without affecting each other; however, in our work this is feasible (see the new results in Q.16). Moreover, our work decomposes the promoters into functionally insulated promoter cores and operators and proves that they can be used to predictably design new promoters. In this way, even a small number of insulated elements can be reused to create a substantial diversity in transcriptional regulation. To our knowledge, our work is the first study that achieves this. We have included these results and discussion into the revised manuscript.

As well as this overall major point, I have a few minor points as well – with points 6 and 7 being particularly important.

Question 6:

1. Line 103 – there should be a space between numbers and units – e.g. 18 bp, or 10 mM. This needs to be fixed throughout the manuscript and supplementary information.

Response:

We have fixed this issue.

Question 7:

2. Line 113 – a word is missing here. Possibly ‘T7’

Response:

We have fixed it.

Question 8:

3. Line 127 – please add more description as to how saturation mutagenesis was done.

Response:

A sentence for further explanation has been added to the main text and the corresponding Methods section has been expanded for detailed description as well.

Question 9:

4. Line 165 – the information in the sentence beginning on this line should be put towards the beginning of the section and not within it.

Response:

We have done it as the reviewer suggested.

Question 10:

5. Lines 174 to 178 – this text seems a bit like it has been taken from a PhD thesis and isn't part of the paper. It refers to 'chapters' for example.

Response:

We agree with the reviewer and have removed this paragraph.

Question 11:

6. Line 200 – The slight cooperative effect with T7 polymerase may be explained by the negative effect it has at high-levels on the growth of E.coli. If T7 polymerase is expressed at high levels or if it activates high-levels of GFP expression from a strong version of a T7 promoter, it can burden the cells, slow the growth rate of the E.coli down and as a result, GFP per cell increases because the cell division rate lowers (lowering the GFP dilution rate). This is a published phenomenon noticed first by Lingchong You's group: Tan, C., Marguet, P. & You, L. Emergent bistability by a growth-modulating positive feedback circuit. *Nat Chem Biol* 5, 842-848 (2009).

Response:

We thank the reviewer for this helpful comment. In our work the experimental characterization of parts and circuits were conducted using flow cytometry, as reported previously⁷. Before cytometry recording, the samples were sufficiently diluted to avoid the saturation of cell flow rate on cytometer. This enables us to quantify the toxicity/burden effect of gene expression on cell growth using the “cell event number/ten seconds” (renamed as “relative growth rate”). We have re-examined the cytometry data of Fig. 3c-d and obtained results for the expression of T7 RNAP and σ^{ECF11} , as shown below:

The horizontal axis denotes the expression level of T7 RNAP/ σ^{ECF11} , while the left vertical axis denotes the cell number. As we can see, the cell growth is insensitive to the expression level of T7 RNAP/ σ^{ECF11} , which means the toxicity/burden effect of these activators on cell growth is negligible in our experiments. Moreover, the hypothesis of “cell growth retardation” cannot explain the discernible sigmoid feature when the expression level is rather low (the bottom-left corner of Fig. 3d). Therefore, the slight cooperativity of T7-dependent transcriptional activation should not be due

to the expression of T7 RNAP. We have included these results and discussions into Supplementary Information (Supplementary Fig. 3) and main text, respectively.

Question 12:

7. *Considering the above point, I would like to also be assured that the various activators/repressors and operator/promoters constructed in this manuscript don't negatively affect the growth rate of E.coli either through burden or off-target effects. The way the data are presented here in the paper makes it impossible to determine if there are any changes to E.coli growth when the promoters are used. Changes to E.coli growth rates will have important downstream consequences such as altering the dilution rate of proteins in gene networks. It is well established that T7 polymerase is challenging for E.coli cells due to driving extraordinarily high gene expression levels and because it transcribes RNA at a rate approximately 3 times faster than normal E.coli RNA polymerase, causing significant stress to the cell and decoupling transcription from translation.*

Response:

In fact, we have paid a lot of attention to the issue of growth burden and cytotoxicity throughout our study. First, all the parts used in this study has been carefully selected according to our experience. For example, the SP6 RNAP was not used because it significantly inhibited *E.coli* growth even with very weak RBS (verified by RBS Calculator). Second, the RBS of each activator/repressor has been carefully fine-tuned to avoid growth burden and cytotoxicity. One example is T7 RNAP, whose RBS was tuned to be very weak (see Supplementary Table 7 of Supplementary Information). Third, the experimental procedure for part/circuit characterization was carefully designed, as recently reported⁷. In this procedure, the cell cultures were almost 1.4×10^5 -fold diluted to provide sufficient growth capacity for cells to amplify the negative effect of parts/circuits on their growth before cytometry recording. We did not observe any significant burden or cytotoxicity exerted by our parts/circuits. These negative effects would probably occur when the circuits are scale-up⁹.

Question 13:

8. *Line 260 – is the model also accurate for the other activators? (the ones derived from other ECFs and T7 polymerase)*

Response:

The accuracy of model prediction for other activators was not systematically examined in our study, but we did probe it for two T7-like activators (T3 and gh-1 RNAPs). The response functions of cI434-based NOT gates using T3 and gh-1 RNAPs as activators can be precisely predicted by Equation (3) (Supplementary Fig. 4), implying that our model might be also accurate for activators besides T7 RNAP and σ^{ECF11} .

Question 14:

9. *Line 277 – what are the black arrows pointing to in Figure 4B and 4C. Are they necessary?*

Response:

They were used for explaining the numbers above each column. We have removed them for clarity.

Question 15:

10. Line 302 – “one-step to reach” – I’ve never heard of this phrase and I don’t think it can be justified as there are more than one steps involved. Considering an alternative phrase would be wise.

Response:

We have replaced this phrase with “mix-and-match”, inspired by the reviewer’s comments in Q.1.

Question 16:

11. Line 328 – in this section (everything for Figure 6) why choose to build a genetic circuit that only uses 1 of the promoters? Building one with several of the different new designed promoters would’ve been better. The work in this section, while very good is actually much more dependent on the performance characteristics of the pTAC promoters, rather than those developed in this study.

Response:

We thank the reviewer for this insightful comment which led us to improve the data quality. We have experimentally adopted five new promoters to replace the wild-type T7 promoter in Fig. 6f, left panel. This yielded five new IFFL networks. All of their response curves could be precisely predicted, as shown below:

Moreover, these response curves appear to have different peak heights but basically the same peak positions. This is actually the advantage of circuit design using insulated transcriptional elements which allows separate tuning of the peak’s height and position. These results and discussions have been added to the manuscript.

Question 17:

12. Line 419 – please insert a relevant citation for “part-mining”

Response:

We have done it as the reviewer suggested.

Question 18:

13. Line 444 – another solution would be to characterise circuits/promoters using RNAseq which would identify unexpected cryptic promoters.

Response:

We have added this point to the text.

- 1 Kukarin, A., Rong, M. & McAllister, W. T. Exposure of T7 RNA polymerase to the isolated binding region of the promoter allows transcription from a single-stranded template. *J Biol Chem* **278**, 2419-2424, doi:10.1074/jbc.M210058200 (2003).
- 2 Kim, S. *et al.* Probing allostery through DNA. *Science* **339**, 816-819, doi:10.1126/science.1229223 (2013).
- 3 Skinner, G. M., Baumann, C. G., Quinn, D. M., Molloy, J. E. & Hoggett, J. G. Promoter binding, initiation, and elongation by bacteriophage T7 RNA polymerase. A single-molecule view of the transcription cycle. *J Biol Chem* **279**, 3239-3244, doi:10.1074/jbc.M310471200 (2004).
- 4 Gong, P. & Martin, C. T. Mechanism of instability in abortive cycling by T7 RNA polymerase. *J Biol Chem* **281**, 23533-23544, doi:10.1074/jbc.M604023200 (2006).
- 5 Lou, C., Stanton, B., Chen, Y. J., Munsky, B. & Voigt, C. A. Ribozyme-based insulator parts buffer synthetic circuits from genetic context. *Nat Biotechnol* **30**, 1137-1142, doi:10.1038/nbt.2401 (2012).
- 6 Nielsen, A. A. *et al.* Genetic circuit design automation. *Science* **352**, aac7341, doi:10.1126/science.aac7341 (2016).
- 7 Zhang, H. M. *et al.* Measurements of Gene Expression at Steady State Improve the Predictability of Part Assembly. *ACS Synth Biol* **5**, 269-273, doi:10.1021/acssynbio.5b00156 (2016).
- 8 Zhao, H. *et al.* Novel T7-like expression systems used for Halomonas. *Metab Eng* **39**, 128-140, doi:10.1016/j.ymben.2016.11.007 (2017).
- 9 Ceroni, F., Algar, R., Stan, G. B. & Ellis, T. Quantifying cellular capacity identifies gene expression designs with reduced burden. *Nat Methods* **12**, 415-418, doi:10.1038/nmeth.3339 (2015).

Reviewers' Comments:

Reviewer #1:

Remarks to the Author:

The authors have addressed major concerns that were raised in the last review and have added many details and experiments to their manuscript that significantly improve their paper. I have no further comments for this manuscript.

Reviewer #2:

Remarks to the Author:

Having looked through the revised manuscript and the rebuttal to reviewer responses I am happy for the publication of this paper to proceed. The authors have addressed all the issues I raised and have added some very impressive further data too.